# ReMaX: Relaxing for Better Training on Efficient Panoptic Segmentation

Shuyang Sun[1]* Weijun Wang[2] Qihang Yu† Andrew Howard[2] Philip Torr[1] Liang-Chieh Chen†
[1]University of Oxford    [2]Google Research

## Abstract

This paper presents a new mechanism to facilitate the training of mask transformers for efficient panoptic segmentation, democratizing its deployment. We observe that due to the high complexity in the training objective of panoptic segmentation, it will inevitably lead to much higher penalization on false positive. Such unbalanced loss makes the training process of the end-to-end mask-transformer based architectures difficult, especially for efficient models. In this paper, we present ReMaX that adds relaxation to mask predictions and class predictions during the training phase for panoptic segmentation. We demonstrate that via these simple relaxation techniques during training, our model can be consistently improved by a clear margin **without** any extra computational cost on inference. By combining our method with efficient backbones like MobileNetV3-Small, our method achieves new state-of-the-art results for efficient panoptic segmentation on COCO, ADE20K and Cityscapes. Code and pre-trained checkpoints will be available at https://github.com/google-research/deeplab2.

## 1 Introduction

Panoptic segmentation [36] aims to provide a holistic scene understanding [63] by unifying instance segmentation [21] and semantic segmentation [24]. The comprehensive understanding of the scene is obtained by assigning each pixel a label, encoding both semantic class and instance identity. Prior works adopt separate segmentation modules, specific to instance and semantic segmentation, followed by another fusion module to resolve the discrepancy [71, 12, 35, 70, 53, 42]. More recently, thanks to the transformer architecture [64, 4], mask transformers [66, 13, 76, 43, 72, 14, 73] are proposed for end-to-end panoptic segmentation by directly predicting class-labeled masks.

Although the definition of panoptic segmentation only permits each pixel to be associated with just one mask entity, some recent mask transformer-based methods [13, 76, 14, 40] apply sigmoid cross-entropy loss (*i.e.*, not enforcing a single prediction via softmax cross-entropy loss) for mask supervision. This allows each pixel to be associated with multiple mask predictions, leading to an extremely unbalanced loss during training. As shown in Figure 1, when using the sigmoid cross-entropy loss to supervise the mask branch, the false-positive (FP) loss can be even $10^3 \times$ larger than the false-negative (FN) loss. Surprisingly, such unbalanced loss leads to better results than using softmax cross-entropy, which indicates that the gradients produced by the FP loss are still helpful for better performance.

However, the radical imbalance in the losses makes it difficult for the network to produce confident predictions, especially for efficient backbones [28, 57, 27], as they tend to make more mistakes given the smaller model size. Meanwhile, the training process will also become unstable due to the large scale loss fluctuation. To address this issue, recent approaches [4, 13, 14, 40] need to carefully clip

---

*Work done during internship at Google Research. Correspondence to: kevinsun@robots.ox.ac.uk
†Work done while at Google.

37th Conference on Neural Information Processing Systems (NeurIPS 2023).

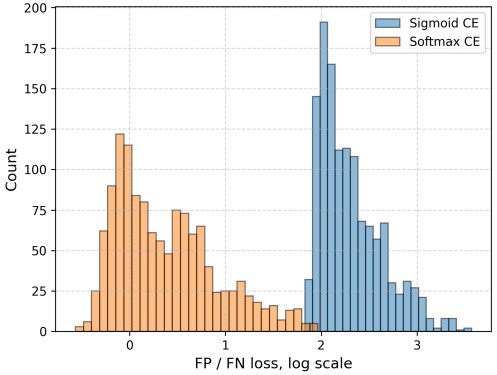

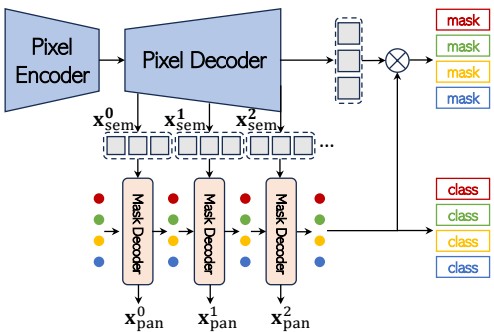

Figure 1: **The histogram shows the ratio of false positives to false negatives for the cross-entropy loss, on a logarithmic scale**. When using sigmoid as the activation function, the false positive loss is always over $100\times$ greater than the false negative, making the total loss to be extremely unbalanced.

Figure 2: **The overall pipeline** for mask-transformers. $\otimes$ represents the matrix multiplication. Here $\mathbf{x}_{\text{pan}}^i$ and $\mathbf{x}_{\text{sem}}^i$ represent the input features of the $i^{th}$ stage for panoptic and semantic heads respectively.

the gradients during training to a very small value like 0.01; otherwise, the loss would explode and the training would collapse. In this way, the convergence of the network will also be slower. A natural question thus emerges: *Is there a way to keep those positive gradients, while better stabilizing the training of the network?*

To deal with the aforementioned conflicts in the learning objectives, one naïve solution is to apply *weighted* sigmoid cross entropy loss during training. However, simply applying the hand-crafted weights would equivalently scale the losses for all data points, which means those positive and helpful gradients will be also scaled down. Therefore, in this paper, we present a way that can adaptively adjust the loss weights by only adding training-time relaxation to mask-transformers [73, 66, 13, 14, 43, 76]. In particular, we propose two types of relaxation: Relaxation on Masks (ReMask) and Relaxation on Classes (ReClass).

The proposed ReMask is motivated by the observation that semantic segmentation is a relatively easier task than panoptic segmentation, where only the predicted semantic class is required for each pixel without distinguishing between multiple instances of the same class. As a result, semantic segmentation prediction could serve as a coarse-grained task and guide the semantic learning of panoptic segmentation. Specifically, instead of directly learning to predict the panoptic masks, we add another auxiliary branch during training to predict the semantic segmentation outputs for the corresponding image. The panoptic prediction is then calibrated by the semantic segmentation outputs to avoid producing too many false positive predictions. In this way, the network can be penalized less by false positive losses.

The proposed ReClass is motivated by the observation that each predicted mask may potentially contain regions involving multiple classes, especially during the early training stage, although each ground-truth mask and final predicted mask should only contain one target in the mask transformer framework [66]. To account for this discrepancy, we replace the original one-hot class label for each mask with a softened label, allowing the ground-truth labels to have multiple classes. The weights of each class is determined by the overlap of each predicted mask with all ground-truth masks.

By applying such simple techniques for relaxation to the state-of-the-art kMaX-DeepLab [73], our method, called ReMaX, can train the network stably without any gradient-clipping operation with a over $10\times$ greater learning rate than the baseline. Experimental results have shown that our method not only speeds up the training by $3\times$, but also leads to much better results for panoptic segmentation. Overall, ReMaX sets a new state-of-the-art record for efficient panoptic segmentation. Notably, for efficient backbones like MobileNetV3-Small and MobileNetV3-Large [27], our method can outperform the strong baseline by $4.9$ and $5.2$ in PQ on COCO panoptic for short schedule training; while achieves $2.9$ and $2.1$ improvement in PQ for the final results (*i.e.*, long schedules). Meanwhile,

our model with a Axial-ResNet50 (MaX-S) [65] backbone outperforms all state-of-the-art methods with 3× larger backbones like ConvNeXt-L [47] on Cityscapes [17]. Our model can also achieve the state-of-the-art performance when compared with the other state-of-the-art efficient panoptic segmentation architectures like YOSO [29] and MaskConver [55] on COCO [44], ADE20K [77] and Cityscapes [17] for efficient panoptic segmentation.

## 2 Related Work

**Mask Transformers for image segmentation.** Recent advancements in image segmentation has proven that Mask Transformers [66], which predict class-labeled object masks through the Hungarian matching of predicted and ground truth masks using Transformers as task decoders [64, 4], outperform box-based methods [35, 70, 54] that decompose panoptic segmentation into multiple surrogate tasks, such as predicting masks for detected object bounding boxes [23] and fusing instance and semantic segmentation [48, 10] with merging modules [42, 53, 45, 71, 12, 41]. The Mask Transformer based methods rely on converting object queries to mask embedding vectors [32, 62, 67], which are then multiplied with pixel features to generate predicted masks. Other approaches such as Segmenter [59] and MaskFormer [14] have also used mask transformers for semantic segmentation. K-Net [76] proposes dynamic kernels for generating masks. CMT-DeepLab [72] suggests an additional clustering update term to improve transformer's cross-attention. Panoptic Segformer [43] enhances mask transformers with deformable attention [79]. Mask2Former [14] adopts masked-attention, along with other technical improvements such as cascaded transformer decoders [4], deformable attention [79], and uncertainty-based point supervision [37], while kMaX-DeepLab [73] employs k-means cross-attention. OneFormer [31] extends Mask2Former with a multi-task train-once design. Our work builds on top of the modern mask transformer, kMaX-DeepLab [73], and adopts novel relaxation methods to improve model capacity.

The proposed Relaxation on Masks (ReMask) is similar to the masked-attention in Mask2Former [14] and the k-means attention in kMaX-DeepLab [73] in the sense that we also apply pixel-filtering operations to the predicted masks. However, our ReMask operation is fundamentally distinct from theirs in several ways: (1) we learn the threshold used to filter pixels in panoptic mask predictions through a semantic head during training, while both masked-attention [14] and k-means attention [73] use either hard thresholding or argmax operation on pixel-wise confidence for filtering; (2) our approach relaxes the training objective by applying a pixel-wise semantic loss on the semantic mask for ReMask, while they do not have explicit supervision for that purpose; and (3) we demonstrate that ReMask can complement k-means attention in Section 4.

**Acceleration for Mask Transformers for efficient panoptic segmentation.** DETR [4] successfully proves that Transformer-based approaches can be used as decoders for panoptic segmentation, however, it still suffer from the slow training problem which requires over 300 epochs for just one go. Recent works [14, 73, 79, 50] have found that applying locality-enhanced attention mechanism can help to boost the speed of training for instance and panoptic segmentation. Meanwhile, some other works [76, 43, 33] found that by removing the bi-partite matching for *stuff* classes and applying a separate group of mask queries for *stuff* classes can also help to speed up the convergence. Unlike them, which apply architectural level changes to the network, our method only applies training-time relaxation to the framework, which do not introduce any extra cost during testing. Apart from the training acceleration, recent works [26, 29, 12, 55, 51] focus on how to make the system for panoptic segmentation more efficient. However, all these works focus on the modulated architecutral design while our approach focus on the training pipeline, which should be two orthogonal directions.

**Coarse-to-fine refinement for image segmentation.** In the field of computer vision, it is a common practice to learn representations from coarse to fine, particularly in image segmentation. For instance, DeepLab [6, 8] proposes a graph-based approach [38, 7] that gradually refines segmentation results. Recently, transformer-based methods for image and video segmentation such as [66, 14, 76, 69, 43, 20, 74, 78] have also adopted a multi-stage strategy to iteratively improve predicted segmentation outcomes in transformer decoders. The concept of using coarse-grained features (*e.g.*, semantic segmentation) to adjust fine-grained predictions (*e.g.*, instance segmentation) is present in certain existing works, including [11, 2, 3]. However, these approaches can lead to a substantial increase in model size and number of parameters during both training and inference.

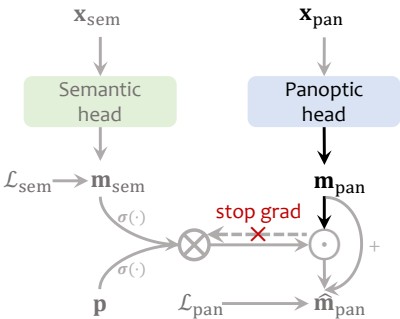

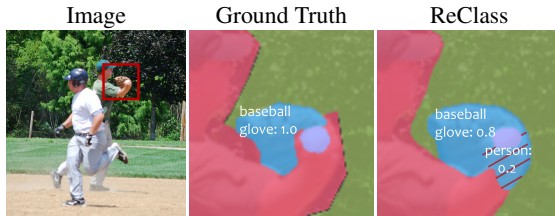

Figure 3: **ReMask Operation.** Modules, representations and operations rendered in gray are ***not*** used in testing. $\otimes$ and $\odot$ represent the matrix multiplication and Hadamard multiplication and + means element-wise sum. The $\times$ symbol and "stop grad" mean that no gradient is flown to $\mathbf{m}_{\text{sem}}$ from $\mathcal{L}_{\text{pan}}$ during training.

Figure 4: **Demonstration on how ReClass works.** We utilize the mask rendered in blue as an example. Our ReClass operation aims to soften the class-wise ground-truth by considering the degree of overlap between the prediction and ground-truth mask. The blue mask intersects with both masks of "baseball glove" and "person", so the final class weights contain both and the activation of "person" in the prediction will no longer be regarded as a false positive case during training.

By contrast, our ReMaX focuses solely on utilizing the coarse-fine hierarchy for relaxation *without* introducing any additional parameters or computational costs during inference.

**Regularization and relaxation techniques.** The proposed Relaxation on Classes (ReClass) involves adjusting label weights based on the prior knowledge of mask overlaps, which is analogous to the re-labeling strategy employed in CutMix-based methods such as [75, 5, 60], as well as label smoothing [61] used in image classification. However, the problem that we are tackling is substantially different from the above label smoothing related methods in image classification. In image classification, especially for large-scale single-class image recognition benchmarks like ImageNet [56], it is unavoidable for images to cover some of the content for other similar classes, and label smoothing is proposed to alleviate such labelling noise into the training process. However, since our approach is designed for Mask Transformers [66, 13, 14, 73, 72] for panoptic segmentation, each image is precisely labelled to pixel-level, there is no such label noise in our dataset. We observe that other than the class prediction, the Mask Transformer approaches also introduce a primary class identification task. The proposal of ReClass operation reduces the complexity for the classification task in Mask Transformers. Prior to the emergence of Mask Transformers, earlier approaches did not encounter this issue as they predicted class labels directly on pixels instead of masks.

## 3  Method

Before delving into the details of our method, we briefly recap the framework of mask transformers [66] for end-to-end panoptic segmentation. Mask Transformers like [66, 14, 76, 69, 43] perform both semantic and instance segmentation on the entire image using a single Transformer-based model. These approaches basically divide the entire framework into 3 parts: a backbone for feature extraction, a pixel decoder with feature pyramid that fuses the feature generated by the backbone, and a transformer mask decoder that translates features from the pixel decoder into panoptic masks and their corresponding class categories.

In the transformer decoder, a set of mask queries is learnt to segment the image into a set of masks by a mask head and their corresponding categories by a classification head. These queries are updated within each transformer decoder (typically, there are at least 6 transformer decoders) by the cross-attention mechanism [64] so that the mask and class predictions are gradually refined. The set of predictions are matched with the ground truth via bipartite matching during training; while these queries will be filtered with different thresholds as post-processing during inference. We follow the same post-processing as kMaX-DeepLab [73].

## 3.1 Relaxation on Masks (ReMask)

The proposed Relaxation on Masks (ReMask) aims to ease the training of panoptic segmentation models. Panoptic segmentation is commonly viewed as a more intricate task than semantic segmentation, since it requires the model to undertake two types of segmentation (namely, instance segmentation and semantic segmentation). In semantic segmentation, all pixels in an image are labeled with their respective class, without distinguishing between multiple instances (*things*) of the same class. As a result, semantic segmentation is regarded as a more coarse-grained task when compared to panoptic segmentation. Current trend in panoptic segmentation is to model *things* and *stuff* in a unified framework and resorts to train both the coarse-grained segmentation task on *stuff* and the more fine-grained segmentation task on *things* together using a stricter composite objective on *things*, which makes the model training more difficult. We thus propose ReMask to exploit an auxiliary semantic segmentation branch to facilitate the training.

**Definition.** Here we first define $H, W$ as the height and width of the feature, $N_Q$ as the number of mask queries. $N_C$ denotes the number of semantic classes for the target dataset, $d_q$ is the number of channels for the query representation, and $d_{\text{sem}}$ is the number of channels for the input of semantic head. As shown in Figure 2 and 3, given a mask representation $\mathbf{x}_{\text{pan}} \in \mathbb{R}^{HW \times N_Q}$, we apply a panoptic mask head to generate panoptic mask logits $\mathbf{m}_{\text{pan}} \in \mathbb{R}^{HW \times N_Q}$. A mask classification head to generate the corresponding classification result $\mathbf{p} \in \mathbb{R}^{N_Q \times N_C}$ is applied for each query representation $\mathbf{q} \in \mathbb{R}^{N_Q \times d_q}$. A semantic head is applied after the semantic feature $\mathbf{x}_{\text{sem}} \in \mathbb{R}^{HW \times d_{\text{sem}}}$ from the pixel decoder to produces a pixel-wise semantic segmentation map $\mathbf{m}_{\text{sem}} \in \mathbb{R}^{HW \times N_C}$ assigning a class label to each pixel. As for the structure for semantic head, we apply an ASPP module [9] and a $1 \times 1$ convolution layer afterwards to transform $d_{\text{sem}}$ channels into $N_C$ channels as the semantic prediction. Note that the whole auxiliary semantic branch will be skipped during inference as shown in Figure 3. Since the channel dimensionality between $\mathbf{m}_{\text{sem}}$ and $\mathbf{m}_{\text{pan}}$ is different, we map the semantic masks into the panoptic space by:

$$\widehat{\mathbf{m}}_{\text{sem}} = \sigma(\mathbf{m}_{\text{sem}})\sigma(\mathbf{p}^{\mathsf{T}}), \tag{1}$$

where $\sigma(\cdot)$ function represents the sigmoid function that normalizes the logits into interval $[0, 1]$. Then we can generate the relaxed panoptic outputs $\widehat{\mathbf{m}}_{\text{pan}}$ in the semantic masking process as follows:

$$\widehat{\mathbf{m}}_{\text{pan}} = \mathbf{m}_{\text{pan}} + (\widehat{\mathbf{m}}_{\text{sem}} \odot \mathbf{m}_{\text{pan}}), \tag{2}$$

where the $\odot$ represents the Hadamard product operation. Through the ReMask operation, the false positive predictions in $\mathbf{m}_{\text{pan}}$ can be suppressed by $\widehat{\mathbf{m}}_{\text{sem}}$, so that during training each relaxed mask query can quickly focus on areas of their corresponding classes. Here we apply identity mapping to keep the original magnitude of $\mathbf{m}_{\text{pan}}$ so that we can remove the semantic branch during testing. This makes ReMask as a complete relaxation technique that does not incur any overhead cost during testing. The re-scaled panoptic outputs $\widehat{\mathbf{m}}_{\text{pan}}$ will be supervised by the losses $\mathcal{L}_{\text{pan}}$.

**Stop gradient for a simpler objective to $\widehat{\mathbf{m}}_{\text{sem}}$.** In order to prevent the losses designed for panoptic segmentation from affecting the parameters in the semantic head, we halt the gradient flow to $\mathbf{m}_{\text{sem}}$, as illustrated in Figure 3. This means that the semantic head is solely supervised by a semantic loss $\mathcal{L}_{\text{sem}}$, so that it can focus on the objective of semantic segmentation, which is a less complex task.

**How does ReMask work?** As defined above, there are two factors that ReMask operation helps training, (1) the Hadamard product operation between the semantic outputs and the panoptic outputs that helps to suppress the false positive loss; and (2) the *relaxation* on training objectives that trains the entire network simultaneously with consistent (*coarse-grained*) semantic predictions. Since the semantic masking can also enhance the locality of the transformer decoder like [14, 73], we conducted experiments by replacing $\mathbf{m}_{\text{sem}}$ with ground truth semantic masks to determine whether it is the training relaxation or the local enhancement that improves the training. When $\mathbf{m}_{\text{sem}}$ is assigned with ground truth, there will be no $\mathcal{L}_{\text{sem}}$ applied to each stage, so that $\mathbf{m}_{\text{pan}}$ is applied with the most accurate local enhancement. In this way, there are large amount of false positive predictions masked by the ground truth semantic masks, so that the false positive gradient will be greatly reduced. The results will be reported in Section 4. The semantic masking can be viewed as local enhancement as it would suppress the extreme false-positive predictions via a simple masking operation.

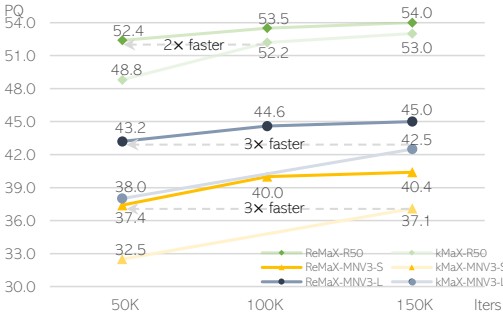

| Method | Backbone | Resolution | FPS | PQ |
|---|---|---|---|---|
| Panoptic-DeepLab [12] | MNV3-L [27] | 641×641 | 26.3 | 30.0 |
| Panoptic-DeepLab [12] | R50 [22] | 641×641 | 20.0 | 35.1 |
| Real-time [26] | R50 [22] | 800×1333 | 15.9 | 37.1 |
| MaskConver [55] | MN-MH [16] | 640×640 | 40.2 | 37.2 |
| MaskFormer [14] | R50 [22] | 800×1333 | 17.6 | 46.5 |
| YOSO [29] | R50 [22] | 800×1333 | 23.6 | 48.4 |
| YOSO [29] | R50 [22] | 512×800 | 45.6 | 46.4 |
| kMaX-DeepLab [73] | R50 [22] | 1281×1281 | 16.3 | 53.0 |
| ReMaX-T† | MNV3-S [27] | 641×641 | 108.7 | 40.4 |
| ReMaX-S† | MNV3-L [27] | 641×641 | 80.9 | 44.6 |
| ReMaX-M‡ | R50 [22] | 641×641 | 51.9 | 49.1 |
| ReMaX-B | R50 [22] | 1281×1281 | 16.3 | **54.2** |

Figure 5: Performance on COCO *val* compared to the baseline kMaX-DeepLab [73]. ReMaX can lead to 3× faster convergence compared to the baseline, and can improve the baselines by a clear margin. The performance of ResNet-50 can be further improved to 54.2 PQ when the model is trained for 200K iterations.

Table 1: Comparison with other state-of-the-art efficient models ($\geq$ 15 FPS) on COCO *val* set. The Pareto curve is shown in Figure 6 (b). The FPS of all models are evaluated on a NVIDIA V100 GPU with batch size 1. †‡ represent the application of efficient pixel and transformer decoders. Please check the appendix for details.

## 3.2 Relaxation on Classes (ReClass)

Mask Transformers [66, 14, 73, 43] operate under the assumption that each mask prediction corresponds to a single class, and therefore, the ground truth for the classification head are one-hot vectors. However, in practice, each imperfect mask predicted by the model during the training process may intersect with multiple ground truth masks, especially during the early stage of training. As shown in Figure 4, the blue mask, which is the mask prediction, actually covers two classes ("baseball glove" and "person") defined in the ground truth. If the class-wise ground truth only contains the class "baseball glove", the prediction for "person" will be regarded as a false positive case. However, the existence of features of other entities would bring over-penalization that makes the network predictions to be under-confident.

To resolve the above problem, we introduce another relaxation strategy on class logits, namely Class-wise Relaxation (ReClass), that re-assigns the class confidence for the label of each predicted mask according to the overlap between the predicted and ground truth semantic masks. We denote the one-hot class labels as $\mathbf{y}$, the ground truth binary semantic masks as $\mathcal{S} = [\mathbf{s}_0, ..., \mathbf{s}_{HW}] \in \{0, 1\}^{HW \times N_C}$, the supplement class weights is calculated by:

$$\mathbf{y}_m = \frac{\sigma(\mathbf{m}_{\text{pan}})^\intercal \mathcal{S}}{\sum_i^{HW} \mathbf{s}_i},\tag{3}$$

where $\mathbf{y}_m$ denotes the label weighted by the normalized intersections between the predicted and the ground truth masks. With $\mathbf{y}_m$, we further define the final class weight $\widehat{\mathbf{y}} \in [0, 1]^{N_C}$ as follows:

$$\widehat{\mathbf{y}} = \eta \mathbf{y}_m + (1 - \eta \mathbf{y}_m)\mathbf{y},\tag{4}$$

where the $\eta$ denotes the smooth factor for ReClass that controls the degree of the relaxation applying to the classification head.

## 4 Experimental Results

### 4.1 Datasets and Evaluation Metric

Our study of ReMaX involves analyzing its performance on three commonly used image segmentation datasets. **COCO** [44] supports semantic, instance, and panoptic segmentation with 80 "things" and 53 "stuff" categories; **Cityscapes** [17] consists of 8 "things" and 11 "stuff" categories; and **ADE20K** [77] contains 100 "things" and 50 "stuff" categories. We evaluate our method using the Panoptic Quality (PQ) metric defined in [36] (for panoptic segmentation), the Average Precision defined in [44] (for instance segmentation), and the mIoU [19] metric (for semantic segmentation).

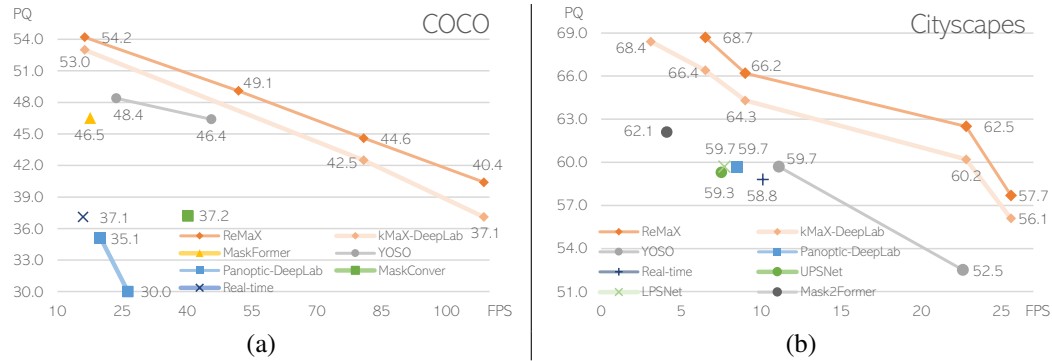

Figure 6: **FPS-PQ Pareto curve** on (a) **COCO Panoptic *val* set** and (b) **Cityscapes *val* set**. Details of the corresponding data points can be found in Table 1 and 9. We compare our method with other state-of-the-art efficient pipelines for panoptic segmentation including kMaX-DeepLab [73], Mask2Former [14], YOSO [29], Panoptic-DeepLab [12], Real-time Panoptic Segmentation [26], UPSNet [70], LPSNet [25], MaskFormer [13], and MaskConver [55].

## 4.2 Results on COCO Panoptic

**Implementation details.** The macro-architecture of ReMaX basically follows kMaX-DeepLab [73], while we incorporate our modules introduced in Section 3 into the corresponding heads. Concretely, we use the *key* in each k-means cross-attention operation as $\mathbf{x}_{\texttt{sem}}$ defined in Figure 3. The semantic head introduced during training consists of an ASPP module [8] and a $1 \times 1$ convolution that outputs $N_C$ number of channels. The specification of models with different size is introduced in the appendix. For other details like post-processing and data preparation, we strictly follow kMaX-DeepLab [73].

**Training details.** We basically follow the training recipe proposed in kMaX-DeepLab [73] but make some changes to the hyper-parameters since we add more relaxation to the network. Here we high-light the necessary and the full training details and specification of our models can be also found in the appendix. The learning rate for the ImageNet-pretrained [56] backbone is multiplied with a smaller learning rate factor 0.1. For training augmentations, we adopt multi-scale training by randomly scaling the input images with a scaling ratio from 0.3 to 1.7 and then cropping it into resolution $1281 \times 1281$. Following [66, 72, 73], we further apply random color jittering [18], and panoptic copy-paste augmentation [33, 58] to train the network. DropPath [30, 39] is applied to the backbone, the transformer decoder. AdamW [34, 49] optimizer is used with weight decay 0.005 for short schedule 50K and 100K with a batch size 64. For long schedule, we set the weight decay to 0.02. The initial learning rate is set to 0.006, which is multiplied by a decay factor of 0.1 when the training reaches 85% and 95% of the total iterations. The entire framework is implemented with DeepLab2 [68] in TensorFlow [1]. Following [66], we apply a PQ-style loss, a Mask-ID cross-entropy loss, and the instance discrimination loss to better learn the feature extracted from the backbone.

For all experiments if not specified, we default to use ResNet-50 as the backbone and apply ReMask to the first 4 stages of transformer decoder. The $\eta$ for ReClass operation is set to 0.1. All models are trained for 27 epochs (*i.e.*, 50K iterations). The loss weight for the auxiliary semantic loss $\mathcal{L}_{\texttt{sem}}$ applied to each stage in the transformer decoder is set to 0.5 and the weights for those loss terms in $\mathcal{L}_{\texttt{pan}}$ are set the same as kMaX-Deeplab[73].

**ReMaX significantly improves the training convergence and outperforms the baseline by a large margin.** As shown in Figure 5, we can see that when training the model under different training schedules 50K, 100K and 150K, our method outperform the baselines by a clear margin for all different schedules. Concretely, ReMaX can outperform the state-of-the-art baseline kMaX-DeepLab by a significant 3.6 PQ when trained under a short-term schedule 50K iterations (27 epochs) for backbone ResNet-50. Notably, our model trained with only 50K iterations performs even better than kMaX-DeepLab [73] trained for the 100K iterations (54 epochs), which means that our model can speed up the training process by approximately $2\times$. We kindly note that the performance of ResNet-50 can be further improved to 54.2 PQ for 200K iterations. ReMaX works very well with

| Activation | w/ ReMaX? | w/ grad-clip? | PQ |
|---|---|---|---|
| softmax | ✗ | ✗ | 48.8 |
| softmax | ✓ | ✗ | 49.5 |
| sigmoid | ✗ | ✗ | 50.4 |
| sigmoid | ✗ | ✓ | 51.2 |
| sigmoid | ✓ | ✗ | **52.4** |

Table 2: **The impact of activation function and gradient clipping.**

| #ReMasks | 0 | 2 | 4 | 6 |
|---|---|---|---|---|
| PQ | 50.4 | 51.9 | **52.4** | 51.5 |

Table 3: **The effect of number of ReMask applied.** ReMaX performs the best when ReMask is applied to the first 4 stages of the transformer decoder.

| $\eta$ | 0 | 0.01 | 0.05 | 0.1 | 0.2 |
|---|---|---|---|---|---|
| PQ | 51.7 | 51.7 | 51.9 | **52.4** | 51.5 |

Table 4: **The impact of different $\eta$ defined in Eq. 4 for ReClass.** Here we observe that the result reaches its peak when $\eta = 0.1$.

| w/ identity mapping? | w/ ReMask in test? | PQ |
|---|---|---|
| ✓ | ✗ | **52.4** |
| ✓ | ✓ | **52.4** |
| ✗ | ✓ | 52.1 |
| ✗ | ✗ | 51.9 |

Table 5: **Effect of applying identity mapping and auxiliary head for ReMask during testing.** Removing the auxiliary semantic head will not lead to performance drop when $\widehat{m}_{pan}$ is applied with identity mapping.

| Method | Backbone | FPS | PQ |
|---|---|---|---|
| MaskFormer [13] | | 17.6 | 46.5 |
| K-Net [76] | | - | 47.1 |
| PanSegFormer [43] | | 7.8 | 49.6 |
| Mask2Former [14] | R50 [22] | 8.6 | 51.9 |
| kMaX-DeepLab [73] | | 26.3 | 53.0 |
| MaskDINO [40] | | 16.8$^\ddagger$ | 53.0 |
| ReMaX | | 26.3$^\dagger$ | **54.2** |

Table 6: **Comparison on COCO val with other models using ResNet-50 as the backbone.** $^\dagger$The FPS here is evaluated under resolution $1200 \times 800$ on V100 and the model is trained for 200K iterations. $^\ddagger$ is evaluated using a A100 GPU.

| w/ stop-grad? | w/ gt? | PQ |
|---|---|---|
| ✓ | ✗ | **52.4** |
| N/A | ✓ | 45.1 |
| ✗ | ✗ | 36.6$^*$ |

Table 7: **The effect of stop gradient and gt-masking.** The denotation *w/ gt?* means whether we use ground-truth semantic masks for $m_{sem}$. $^*$ The result without the stop-gradient operation does not well converge in training.

efficient backbones including MobileNetV3-Small [27] and MobileNetV3-Large [27], which surpass the baseline performance by 4.9 and 5.2 PQ for 50K iterations, and 3.3 and 2.5 PQ respectively for 150K iterations. These results demonstrate that the proposed relaxation can significantly boost the convergence speed, yet can lead to better results when the network is trained under a longer schedule.

**ReMaX *vs.* other state-of-the-art models for efficient panoptic segmentation.** Table 1 and Figure 6 (a) compares our method with other state-of-the-art methods for efficient panoptic segmentation on COCO Panoptic. We present 4 models with different resolution and model capacity, namely ReMaX-Tiny (T), ReMaX-Small (S), ReMaX-Medium (M) and ReMaX-Base (B). Due to the limit of space, the detailed specification of these models is included in the appendix. According to the Pareto curve shown in Figure 6 (a), our approach outperforms the previous state-of-the-art efficient models by a clear margin. Specifically, on COCO Panoptic *val* set, our models achieve 40.4, 44.6, 49.1 and 54.2 PQ with 109, 81, 52 and 16 FPS for ReMaX-T, ReMaX-S, ReMaX-M and ReMaX-B respectively. The speed of these models is evaluated under the resolution $641 \times 641$ except for ReMaX-Base, which is evaluated under resolution $1281 \times 1281$. Meanwhile, as shown in Table 6, our largest model with the backbone ResNet-50 also achieves better performance than the other non-efficient state-of-the-art methods with the same backbone.

**Effect of different activation, and the use of gradient clipping.** Table 2 presents the effect of using different activation function (sigmoid *vs.* softmax) for the Mask-ID cross-entropy loss and the $\sigma(\cdot)$ defined in Eq (1). From the table we observe that ReMask performs better when using sigmoid as the activation function, but our method can get rid of gradient clipping and still get a better result.

**Can we use the ground-truth masks for local enhancement instead of ReMask?** As discussed in Section 3, to figure out whether it is the loss relaxation or the pixel filtering that improves the training, we propose experiments to replace $m_{sem}$ with the ground truth semantic masks during training. When $m_{sem}$ is changed into the ground truth, all positive predictions outside the ground-truth masks will be removed, which means that the false positive loss would be significantly scaled down. The huge drop (52.4 *vs.* 45.1 PQ in Table 7) indicates that the gradients from false positive losses can benefit the final performance. Table 7 also shows that when enabling the gradient flow from the panoptic loss to the semantic predictions, the whole framework cannot converge well and lead to a drastic drop in performance (36.6 PQ). The semantic masks $m_{sem}$ faces a simpler objective (*i.e.* only semantic segmentation) if the gradient flow is halted.

| w/semantic masking (local enhancement) | w/ $L_{sem}$ (loss relaxation) | w/ ReClass ? | Iterations | PQ |
|:---:|:---:|:---:|:---:|:---:|
| × | × | × | 50K | 50.4 |
| × | × | × | 150K | 53.0 |
| × | ✓ | × | 50K | 51.3 |
| × | ✓ | × | 150K | 53.0 |
| ✓ | ✓ | × | 50K | 51.7 |
| ✓ | ✓ | ✓ | 50K | **52.4** |
| ✓ | ✓ | ✓ | 150K | **54.0** |

Table 8: The relative impact of *loss relaxation* and *semantic masking* (local enhancement) on COCO Panoptic *val* set under short (50K) and long (150K) training schedule.

**The relative contribution of loss relaxation and local enhancement in ReMask.** The auxiliary semantic loss term can be viewed as loss relaxation, while the semantic masking branch can be viewed as local enhancement. To disentangle the relative contribution of loss relaxation and local enhancement, we conducted another ablation study that removes the semantic masking branch (the concrete grey arrow right under "stop grad" in Figure 3), which would remove the local enhancement (semantic masking) but keep the auxiliary semantic loss term for loss relaxation. The results are reported in Table 8. The short training schedule of 50K iterations shows that the semantic loss relaxation leads to a 0.9 increase in PQ; while the semantic masking contributes to an additional 0.4 gain in PQ. The long-schedule training (i.e. 150K iterations) demonstrates that semantic masking is critical in ReMask because applying semantic loss relaxation alone without semantic masking does not result in any improvement. In other words, only using semantic loss relaxation may expedite the early stage of training (e.g, 50K iterations), but it fails to improve the ultimate convergence quality.

**The number of mask relaxation.** Table 3 shows the effect of the number of ReMask applied to each stage, from which we can observe that the performance gradually increases and reaches its peak at 52.4 PQ when the number of ReMask is 4, which is also our final setting for all other ablation studies. Using too many ReMask ($> 4$) operations in the network may add too many relaxation to the framework, so that it cannot fit well to the final complex goal for panoptic segmentation.

**ReClass can also help improve the performance for ReMaX.** We investigate ReClass and its hyper-parameter $\eta$ in this part and report the results in Table 4. In Table 4, we ablate 5 different $\eta$ from 0 to 0.2 and find that ReClass performs the best when $\eta = 0.1$, leading to a 0.5 gain compared to the strong baseline. The efficacy of ReClass validates our assumption that each mask may cover regions of multiple classes.

**Effect of the removing auxiliary semantic head for ReMask during testing.** The ReMask operation can be both applied and removed during testing. In Table 5, it shows that the accuracy is comparable under the two settings. In Table 5 we also show the necessity of applying identity mapping to $\mathbf{m}_{\text{pan}}$ during training in order to remove the auxiliary semantic head during testing. Without the identity mapping at training, removing semantic head during testing would lead to $0.5$ drop from $52.4$ (the first row in Table 5) to $51.9$.

### 4.3 Results on Cityscapes

**Implementation details.** Our models are trained using a batch size of 32 on 32 TPUv3 cores, with a total of 60K iterations. The first 5K iterations constitute the warm-up stage, where the learning rate gradually increases from 0 to $3 \times 10^{-3}$. During training, the input images are padded to $1025 \times 2049$ pixels. In addition, we employ a multi-task loss function that includes four loss components with different weights. Specifically, the weights for the PQ-style loss (part of $\mathcal{L}_{\text{pan}}$), auxiliary semantic loss $\mathcal{L}_{\text{sem}}$, mask-id cross-entropy loss (part of $\mathcal{L}_{\text{pan}}$), and instance discrimination loss are set to 3.0, 1.0, 0.3 and 1.0, respectively. To generate feature representations for our model, we use 256 cluster centers and incorporate an extra bottleneck block in the pixel decoder, which produces features with an output stride of 2. These design are basically proposed in kMaX-DeepLab [73] and we simply follow here for fair comparison.

**Results on Cityscapes.** As shown in Table 9 and Figure 6 (b), it shows that our method can achieve even better performance when using a smaller backbone MobileNetV3-Large (62.5 PQ) while the

| Method | Backbone | FPS | PQ |
|---|---|---|---|
| Mask2Former [14] | R50 [22] | 4.1 | 62.1 |
| Panoptic-DeepLab [12] | Xception-71 [15] | 5.7 | 63.0 |
| LPSNet [25] | R50 [22] | 7.7 | 59.7 |
| Panoptic-DeepLab [12] | R50 [22] | 8.5 | 59.7 |
| kMaX-DeepLab [73] | R50 [22] | 9.0 | 64.3 |
| Real-time [26] | R50 [22] | 10.1 | 58.8 |
| YOSO [29] | R50 [22] | 11.1 | 59.7 |
| kMaX-DeepLab [73] | MNV3-L [27] | 22.8 | 60.2 |
| ReMaX | R50 [22] | 9.0 | **65.4** |
| ReMaX | MNV3-L [27] | 22.8 | **62.5** |
| ReMaX | MNV3-S [27] | 25.6 | **57.7** |

Table 9: Cityscapes *val* set results for lightweight backbones. We consider methods without pre-training on extra data like COCO [44] and Mapillary Vistas [52] and test-time augmentation for fair comparison. We evaluate our FPS with resolution $1025 \times 2049$ and a V100 GPU. The FPS for other methods are evaluated using the resolution reported in their original papers.

| Method | Backbone | FPS | #params | PQ |
|---|---|---|---|---|
| Mask2Former [73] | Swin-L[†] [46] | - | 216M | 66.6 |
| kMaX-DeepLab [73] | MaX-S[†] [66] | 6.5 | 74M | 66.4 |
| kMaX-DeepLab [73] | ConvNeXt-L[†] [47] | 3.1 | 232M | 68.4 |
| OneFormer [31] | ConvNeXt-L[†] [47] | - | 220M | 68.5 |
| ReMaX | MaX-S[†] [27] | 6.5 | 74M | **68.7** |

Table 10: Cityscapes *val* set results for larger backbones. [†]Pre-trained on ImageNet-22k.

| Method | Backbone | Resolution | FPS | PQ | mIoU |
|---|---|---|---|---|---|
| MaskFormer [13] |  | 640-2560 | - | 34.7 | - |
| Mask2Former [14] |  | 640-2560 | - | 39.7 | 46.1 |
| YOSO [29] | R50 [22] | 640-2560 | 35.4 | 38.0 | - |
| kMaX-DeepLab [73] |  | 641×641 | 38.7 | 41.5 | 45.0 |
| kMaX-DeepLab [73] |  | 1281×1281 | 14.4 | 42.3 | 45.3 |
| ReMaX | R50 [22] | 641×641 | 38.7 | **41.9** | **45.7** |
| ReMaX |  | 1281×1281 | 14.4 | **43.4** | **46.9** |

Table 11: ADE20K *val* set results. Our FPS is evaluated on a NVIDIA V100 GPU under the corresponding resolution reported in the table.

other methods are based on ResNet-50. Meanwhile, our model with Axial-ResNet-50 (*i.e.*, MaX-S, 74M parameters) as the backbone can outperform the state-of-the-art models [31, 73] with a ConvNeXt-L backbone (> 220M parameters). The Pareto curve in Figure 6 (b) clearly demonstrates the efficacy of our method in terms of speed-accuracy trade-off.

### 4.4   Results on ADE20K

**Implementation details.**   We basically follow the same experimental setup as the COCO dataset, with the exception that we train our model for 100K iterations (54 epochs). In addition, we conduct experiments using input resolutions of $1281 \times 1281$ pixels and $641 \times 641$ respectively. During inference, we process the entire input image as a whole and resize longer side to target size then pad the shorter side. Previous approaches use a sliding window approach, which may require more computational resources, but it is expected to yield better performance in terms of accuracy and detection quality. As for the hyper-parameter for ReMask and ReClass, we used the same setting as what we propose on COCO.

**Results on ADE20K.**   In Table 11, we compared the performance of ReMaX with other methods, using ResNet-50 as the backbone, and found that our model outperforms the baseline model by 1.6 in terms of mIOU, which is a clear margin compared to the baseline, since we do not require any additional computational cost but only the relaxation during training. We also find that our model can surpass the baseline model kMaX-DeepLab by 1.1 in terms of PQ. When comparing with other frameworks that also incorporate ResNet-50 as the backbone, we show that our model is significantly better than Mask2Former and MaskFormer by 3.7 and 8.7 PQ respectively.

## 5   Conclusion

This paper presents a novel approach called ReMaX, comprising two components, ReMask and ReClass, that leads to better training for panoptic segmentation with Mask Transformers. The proposed method is shown to have a significant impact on training speed and final performance, especially for efficient models. In principle, ReMaX has the potential to be generalized to other non-transformer-based panoptic segmentation frameworks as long as it has a panoptic mask representation and a semantic mask representation . In this paper, we mainly verify our method based on state-of-the-art mask transformers [73]. We will further validate the generalization capability of our method in future work. We hope that our work will inspire further investigation in this direction, leading to more efficient and accurate panoptic segmentation models.

**Acknowledgement.** We would like to thank Yukun Zhu, Xuan Yang at Google Research for their kind help and discussion. Shuyang Sun and Philip Torr are supported by the UKRI grant: Turing AI Fellowship EP/W002981/1 and EPSRC/MURI grant: EP/N019474/1. We would also like to thank the Royal Academy of Engineering and FiveAI.

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

# Appendix

## A    Loss Visualization of ReMaX

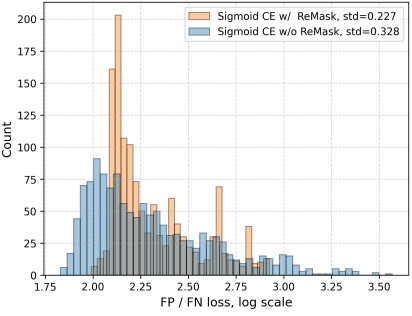

Figure 7: The histogram shows the ratio of false positives to false negatives for the cross-entropy loss, on a logarithmic scale.

| Method | Backbone | #Params | FLOPs | FPS | PQ |
|---|---|---|---|---|---|
| kMaX-DeepLab [73] | ConvNeXt-T$^\dagger$ [66] | 61M | 172G | 21.8 | 55.3 |
| ReMaX | ConvNeXt-T$^\dagger$ [66] | 61M | 172G | 21.8 | **55.9** |
| Mask2Former [14] | Swin-B$^\dagger$ [46] | 107M | 466G | - | 56.4 |
| kMaX-DeepLab [73] | ConvNeXt-S$^\dagger$ [66] | 83M | 251G | 16.5 | 56.3 |
| ReMaX | ConvNeXt-S$^\dagger$ [66] | 83M | 251G | 16.5 | **56.6** |

Table 12: Results for larger models on COCO *val* set. FLOPs and FPS are evaluated with the input size $1200 \times 800$ and a V100 GPU. $\dagger$: ImageNet-22K pretraining.

We visualize the loss applied with ReMask and the loss applied without ReMask in Figure 7, from which we can observe that ReMask can effectively reduce extremely high false positive losses; therefore, our method can stabilize the training of the framework.

## B    Model Specification

| Model | Backbone | Resolution | #Pixel Decoders | #Transformer Decoders | #FLOPs | #Params | FPS |
|---|---|---|---|---|---|---|---|
| ReMaX-T | MNV3-S [27] | $641 \times 641$ | [1, 1, 1, 1] | [1, 1, 1] | 18.8G | 18.6M | 109 |
| ReMaX-S | MNV3-L [27] | $641 \times 641$ | [1, 1, 1, 1] | [1, 1, 1] | 20.9G | 22.0M | 81 |
| ReMaX-M | R50 [22] | $641 \times 641$ | [1, 5, 1, 1] | [1, 1, 1] | 67.8G | 50.8M | 52 |
| ReMaX-B | R50 [22] | $1281 \times 1281$ | [1, 5, 1, 1] | [2, 2, 2] | 294.7G | 56.6M | 26 |

Table 13: Specification of different models in ReMaX family.

We provide the specification of our models and their corresponding number of parameters and FLOPs in Table 13. We kindly note that the numbers of pixel decoders with the format $[\cdot, \cdot, \cdot, \cdot]$ represent the numbers for features with $[\frac{1}{32}, \frac{1}{16}, \frac{1}{8}, \frac{1}{4}]$ times of the input size. We use Axial attention [65] for all feature maps with resolution $\frac{1}{32}, \frac{1}{16}$ of the input size, and regular bottleneck residual blocks [22] for the rest. The denotation $[\cdot, \cdot, \cdot]$ for the transformer decoders represents the numbers for resolution of $[\frac{1}{16}, \frac{1}{8}, \frac{1}{4}]$ times of the input size.

## C    Performance for Larger Models

We also validate the performance of ReMaX for larger models *e.g.* ConvNeXt-Tiny (T) and ConvNeXt-Small (S). From Table 12 we can find that ReMaX can achieve better results compared to the baseline kMaX-DeepLab [73] and Mask2Former [14]. However, the improvement of ReMaX gets saturated when the numbers become high. Notably, when using ConvNeXt-T backbone, ReMaX can lead to 0.6 PQ increase over kMaX-DeepLab, while incurring ***no extra*** computational cost during inference. The improvement is noticeable, as kMaX-DeepLab only further improves 1.0 PQ by using ConvNeXt-S backbone, at the cost of extra 36% more parameters (22M) and 46% more FLOPs (79G).

## D    Limitations

Since we implement our method in TensorFlow, the baselines we can build upon is limited. We have re-implemented ReMaX in PyTorch and applied it with Mask2former [14]. The result is

| Methods | Epochs | PQ |
|---|---|---|
| Mask2former | 24 | 48.36 |
| Mask2former + ReMaX | 24 | 50.24 |

Table 14: ReMaX is effective on Mask2Former [14] on COCO Panoptic *val* set.

reported in table 14. Due to the time limit, we did not reproduce the originally reported Mask2Former results by fully exploring all the hyper-parameters. However, the table above shows that based on the same Mask2Former baseline ReMaX boost the overall accuracy. This could demonstrate that ReMaX is also effective for other segmentation frameworks like Mask2former. In future work, we need to validate ReMaX based on more frameworks such as YOSO in PyTorch to demonstrate its effectiveness. Meanwhile, ReClass measures the weight of each class according to the size of each mask, which may not be accurate and can be further improved in the future.

# E  Boarder Impact

Our method can help better train models for efficient panoptic segmentation. It can also be used to develop new applications in areas such as autonomous driving, robotics, and augmented reality. For example, in autonomous driving, efficient panoptic segmentation can be used to identify and track other vehicles, pedestrians, and obstacles on the road. This information can be used to help the car navigate safely. In robotics, efficient panoptic segmentation can be used to help robots understand their surroundings and avoid obstacles. This information can be used to help robots perform tasks such as picking and placing objects or navigating through cluttered environments. In augmented reality, efficient panoptic segmentation can be used to overlay digital information on top of the real world. This information can be used to provide users with information about their surroundings or to help them with tasks such as finding their way around a new city. Overall, our method can be used to boost a variety of applications in the field of computer vision and robotics.

