# OpenReview forum: "ReMaX: Relaxing for Better Training on Efficient Panoptic Segmentation"
_NeurIPS.cc/2023/Conference — NeurIPS 2023 poster_

### Official Review · Reviewer_EbKy · 2023-06-30

**Soundness:** 3 good
**Presentation:** 4 excellent
**Contribution:** 2 fair
**Rating:** 6
**Confidence:** 5

**Summary:**

This paper presented a new mechanism to train efficient panoptic segmentation frameworks, which adds relaxation to mask predictions and class prediction for panoptic segmentation. In experiments, the authors demonstrated that the relaxation techniques can consistently improve panoptic segmentation frameworks.

**Strengths:**

1. The results are impressive, especially, with R50 backbone, ReMax-M achieves 49.1 PQ and 51.9 FPS on COCO dataset.
2. This paper is well-written and the motivation is clear.


**Weaknesses:**

1. The generalization of the proposed training mechanism should be demonstrated.
2. Some experimental results are unconvincing.

Pleas see questions and limitations for details.

**Questions:**

1. Why can the loss function with semantic masks be a relaxation that helps training? Training with semantic masks has been demonstrated to improve the instance segmentation in many frameworks such as CondInst. Therefore, what is the main contribution or the novelty of the proposed ReMask should be carefully discussed.
2. In Table 6, the results of MaskDINO and ReMaX should be tested with same GPU. When the GPU is V100, ReMax achieves 16.3 FPS (from Table 1), but MaskDINO achieves 16.8 FPS.

Typo:
1. Line 71: "... like YOSO [26] and MaskConver [26] ... "

**Limitations:**

1. The contributions of this paper may be limited. For the segmentation framework, this paper directly uses the kMax-DeepLab. For the ReMask technique, it has been demonstrated in instance segmentation. For the ReClass technique, the effect of applying it or not is not shown in experiments.

2. The generalization of the proposed method should be demonstrated based on more frameworks such as Mask2Former, YOSO, and Panoptic DeepLab, not only kMax-Deeplab.

---

> ### Author Rebuttal · Authors · 2023-08-09
>
> We are grateful to the reviewer for the insightful feedback. We hope that the subsequent response will address the concerns voiced in the review. We thank the reviewer for pointing out the typo, which we will fix in the final revision.
>
> | *w/* semantic masking? | *w/* $\mathcal{L}\_{sem}$? | *w/* ReClass? | Iterations | PQ |
> | :--------------------: | :------------------------: | :-: | :-: |  :-: |
> | | | | 50K | 50.4 |
> | | | | 150K | 53.0 |
> | | &#x2611; | | 50K | 51.3 |
> | | &#x2611; | | 150K | 53.0 |
> |         &#x2611;       | &#x2611; | | 50K | 51.7 |
> |         &#x2611;       | &#x2611; | &#x2611; | 50K | 52.4 |
> |         &#x2611;       | &#x2611; | &#x2611; | 150K | 54.0 |
>
> *Q1. Training with semantic masks has been demonstrated to improve instance segmentation in many frameworks such as CondInst. Therefore, what is the main contribution or the novelty of the proposed ReMask should be carefully discussed.*
>
> We appreciate the insightful suggestion. We echo that incorporating semantic loss such as CondInst [A1] and YOLACT [A2] can indeed enhance the performance of instance segmentation. This is also evident from our experiments, as demonstrated in the table above.
>
> However, we respectfully point out that our approach differs from the application of loss in [A1] and [A2] in the following aspects:
>
> 1. **The key of ReMask is semantic masking instead of purely semantic loss.**
>    As demonstrated in the above table, the direct application of semantic loss, without semantic masking, does not result in any improvement for long-schedule training (i.e., 150K iterations). It's only when semantic masking is implemented that the network tends toward better convergence, a central aspect of our methodology. While the exclusive use of semantic loss may expedite the initial stages of the training process (e.g, 50K iterations), it fails to improve the ultimate convergence quality.
>
> 2. **The ReMask is applied along with mask transformers while CondInst and YOLACT are not.**
>    We kindly argue that our method is applied with mask transformers while CondInst and YOLACT are conventional segmentation frameworks.
>
> 3. The meaning of relaxation is two-fold:
>
>    (1) We posit that compared to panoptic segmentation, semantic segmentation presents a less challenging task. As such, employing semantic prediction to fine-tune the results of panoptic segmentation can be viewed as a form of relaxation strategy.
>
>    (2) The ReClass process alters the initial one-hot label into a softer tensor, thereby easing the intensity of the strict supervision and can be regarded as a way of relaxation.
>
> We would add the above discussion in the revised paper and carefully clarify the difference between ours and the related papers.
>
> ---
>
> *Q2. In Table 6, the results of MaskDINO and ReMaX should be tested with the same GPU. When the GPU is V100, ReMax achieves 16.3 FPS (from Table 1), but MaskDINO achieves 16.8 FPS.*
>
> We kindly note that 16.8 FPS (typo, should be 14.8 in the original paper) of Mask DINO is evaluated on **`A100`**, while our 16.3 FPS is evaluated on **`V100`** and a higher resolution 1281x1281. Here we report the detailed FPS below:
>
> | Method | GPU | Resolution | FPS |
> | :----: | :----: | :-: | :-: |
> | Mask DINO | A100 | Not reported | 14.8 |
> | Mask DINO | V100 | 1200x800 $^\dagger$ | 10.9 |
> | Ours | V100 | 1200x800 | 26.3 |
> | Ours | V100 | 1281x1281 | 16.3 |
>
> For fair comparison, as shown in the table above, ours is about 2x faster than the Mask DINO model for 1200x800 resolution on V100. We will revise the manuscript accordingly to further clarify the confusion.
> $^\dagger$ We recompute the FPS of Mask DINO on our own device.
>
> ---
>
> *L1.  For the ReMask technique, it has been demonstrated in instance segmentation. For the ReClass technique, the effect of applying it or not is not shown in experiments.*
>
> We kindly note that in Table 4 of the manuscript, we have reported that the application of ReClass results in a **`0.7`** PQ increase when comparing the second column ($\eta=0$) with the fifth column ($\eta=0.1$). For distinctions between ReMask and prior methods [A1] and [A2], please refer to the previous response.
>
> ---
>
> *L2. The generalization of the proposed method should be demonstrated based on more frameworks such as Mask2Former, YOSO, and Panoptic DeepLab, not only kMax-Deeplab.*
>
> This is a good suggestion. We have re-implemented ReMaX for Mask2former in PyTorch and reported the results in the table below.
> | Method | Epochs | PQ |
> | :----: | :----: | :-:|
> | Mask2former | 24 | 48.36 |
> | Mask2former + ReMaX | 24 | 50.24 |
>
> Due to the time limit, we did not reproduce the originally reported Mask2Former results by fully exploring all the hyper-parameters. However, the table above shows that based on the same Mask2Former baseline ReMaX boost its accuracy. This could demonstrate that ReMaX is also effective for other segmentation frameworks like Mask2former. We will add such results on more baseline models like Mask2Former and YOSO to the final paper.
>
> ---
>
> ### Reference
> [A1]: Tian Z, Shen C, Chen H. Conditional convolutions for instance segmentation. ECCV 2020.
>
> [A2]: Bolya D, Zhou C, Xiao F, et al. Yolact: Real-time instance segmentation. ICCV 2019.

---

> > ### Comment · Reviewer_EbKy · 2023-08-18
> >
> > Thanks for the feedback. The response has solved my concerns. I will raise my score.

---

> > > ### Author Response · Authors · 2023-08-21
> > > **Thank you for the kind feedback**
> > >
> > > We are happy to see the above feedback solved the concerns of the reviewer. We thank the reviewer for all the constructive comments.

---

### Official Review · Reviewer_Mpjf · 2023-07-02

**Soundness:** 3 good
**Presentation:** 3 good
**Contribution:** 3 good
**Rating:** 6
**Confidence:** 4

**Summary:**

This paper presents a relaxation technique for training Efficient Panoptic Segmentation models called ReMaX. Based on the observation of much higher false positive penalisation in training panoptic segmentation models, it introduces two relaxation designs, ReMask and ReClass. Results are reported on COCO, ADE20K and Cityscapes datasets.

**Strengths:**

1. The motivation of this paper is clear, and the finding is interesting.
2. The proposed method is reasonable and fits well with the motivation.
3. Using soft semantic segmentation prediction for relaxing the training of mask transformers makes sense to me.
4. Consistent improvements are obtained with the proposed method, and ablations are thorough. According to the experiments, the proposed method can work well with multiple existing approaches.

**Weaknesses:**

1. The design seems coupled with mask transformers. It may not be generalised to all efficient panoptic segmentation models.

**Questions:**

About the ReClass operation. What if the overlapped objects belong to the same category? For instance, two 'persons' in Fig. 3.

**Limitations:**

Error bars are not reported.

---

> ### Author Rebuttal · Authors · 2023-08-09
>
> We deeply appreciate the reviewer for the recognition of our paper.
>
> *W1. The design seems coupled with mask transformers. It may not be generalised to all efficient panoptic segmentation models.*
>
> We thank the reviewer for the suggestion and agree that it is interesting to further explore the effectiveness of our method with other segmentation frameworks. In principle, ReMaX has the potential to be generalized to other non-transformer-based panoptic segmentation frameworks with a panoptic mask representation $m\_{pan}$ and a semantic mask representation $m\_{sem}$. But it is beyond the scope of this work and rebuttal. We currently mainly explore our method in kMaX-DeepLab (most recent state-of-the-art), and have also quickly experimented with Mask2Former in this rebuttal. We will further validate the generalizability and effectiveness of our method in future work.
>
> ---
>
> *Q1. About the ReClass operation. What if the overlapped objects belong to the same category? For instance, two 'persons' in Fig. 3.*
>
> This is a good question. Since different instances with the same category belong to the same semantic mask, the class labels for their instance masks **will not change**.
>
> ---
>
> *L1. Error bar not provided.*
>
> Thanks for the suggestion. We will add it to the revised paper.

---

> > ### Comment · Reviewer_Mpjf · 2023-08-19
> >
> > Thanks for the feedback. The proposed method is technically sound, but the impact may be limited by only exploring the method based on one panoptic segmentation framework (kMaX-DeepLab). In the rebuttal, the authors provide an analysis of the possibilities of applying the techniques to other frameworks and mention a quick trial on Mask2Former. However, I don't find the results in their rebuttal.

---

> > > ### Comment · Reviewer_Mpjf · 2023-08-19
> > >
> > > I find the results within the reply to reviewer EbKy. I will consider increasing my rating.

---

> > > > ### Author Response · Authors · 2023-08-21
> > > > **Thank you for your kind feedback**
> > > >
> > > > We are sorry for the confusion that the reviewer found it hard for him/her to find the corresponding results we provided in the rebuttal. Here we kindly point out that it is the last table we showed to reviewer EbKy.
> > > >
> > > > Thank you for all the constructive comments! We are glad to hear that the reviewer is considering raising the score after reading our rebuttal.

---

### Official Review · Reviewer_c7F1 · 2023-07-05

**Soundness:** 4 excellent
**Presentation:** 4 excellent
**Contribution:** 4 excellent
**Rating:** 8
**Confidence:** 4

**Summary:**

This paper shows that the existing sota panoptic segmentation methods have an unbalanced loss ( excessively large false-positive loss due to the use of the sigmoid function). The authors designed two relaxation mechanisms to relax the supervision at the mask and class levels, thereby improving training efficiency and improve accuracy.

**Strengths:**

1. This paper reveals the problem of excessive false-positive loss caused by the use of the sigmoid function in current transformer-based panoptic segmentation models and proves that false-positive loss is also helpful for training and proper relaxation of constraints can improve training efficiency, which is instructive for the community.
2. Both approaches to relaxation in the paper (ReMask and ReClass) design are interesting and effective.
3. The proposed ReMax is excellent in training efficiency, inference efficiency, and accuracy.

**Weaknesses:**

1. On the one hand, excessive false-postive loss affects training efficiency, but on the other hand, false-postive loss also benefits the final result. The author provides only an empirical solution, lacking more in-depth discussions. I wonder about the qualitative analysis of the impact of different scales of false-positive loss on the final result.

**Questions:**

see weakness

**Limitations:**

No applicable

---

> ### Author Rebuttal · Authors · 2023-08-09
>
> We deeply appreciate the reviewer for the recognition of our paper.
>
> *Q. Qualitative analysis of the impact of different scales of false-positive loss on the final result.*
>
> Well spotted! We experimented with various methods to adjust the scales of FP/FN losses.
> Please refer to the table below for these results, where the reported result for FP loss scaling represents the most optimal scaling factor we have tried.
> Overall, as summarized in line 39-44, we observed that equivalently scaling the magnitude of false-positive losses for all examples doesn't enhance performance.
> The crux of ReMaX's effectiveness lies in its ability to dynamically filter out extreme losses, drawing parallels with gradient-clipping.
>
>
> | Loss-scale Method |   PQ  |
> | :---------------: |   :-  |
> | baseline          |  50.4 |
> | w/ FP loss scale $\downarrow$  |  50.4 |
> | w/ FN loss scale $\uparrow$  |  50.9 |
> | w/ Grad-clip  |  51.2 |
> | w/ ReMask  |  51.7 |
> | w/ ReMask+ReClass  |  52.4 |
>
> Based on the above table, it's evident that holistically scaling down the false-positive loss doesn't yield a performance boost.
> In fact, we've tested various scaling factors to holistically scale down the false-positive loss, yet none contributed to better performance.
> The result for FP loss scaling reported here represents the most optimal scaling factor we identified.

---

### Official Review · Reviewer_8sND · 2023-07-06

**Soundness:** 3 good
**Presentation:** 1 poor
**Contribution:** 2 fair
**Rating:** 6
**Confidence:** 5

**Summary:**

The manuscript presents two novel heuristics for training efficient panoptic models based on mask-level recognition and pixel-to-mask assignment.

The first heuristics affects pixel-to-mask assignment and is referred to as ReMask. ReMask has been designed to balance the overwhelming contribution of false positive mask assignments to the mask-assignment loss by leveraging an independent semantic prediction head. In particular, the authors suppress the pixel assignment towards masks that get recognized into classes that are inconsistent with local semantic predictions. However, Table 7 suggests that most of the improvement does not stem from training relaxation and that the benefits may be caused by enhanced locality of the recognition process through Lsem.

The second heuristics affects mask-level recognition and is referred to as ReClass. ReClass changes the classification targets of the predicted masks from one-hot winner-takes-all to mixtures of one-hot assignments of all incident ground-truth masks. It appears that the authors conjecture that ReClass contributes to the convergence speed by reducing the penalty of inaccurate masks during early training.


**Strengths:**

S1. Panoptic segmentation is an important computer vision task with many applications.

S2. State-of-the-art performance among approaches based on mid-range backbones (RN50, MNV3).

S3. Ablations and validations suggest that the proposed heuristics contribute significant performance improvements.

S4. The proposed heuristics can be removed during inference; this results in competitive inference speeds.

**Weaknesses:**

W1. the manuscript requires non-linear reading effort:
* lines 161-181 start to make sense only after reading the equations
* equations for x_pan and x_sem are missing (they should start from shared features)
* d_pan, d_sem, N_Q and N_C should be defined before use.

W2. a bird's-eye view figure is missing (example: Fig.2 in [10]).

W3. many small details:
* l161: it appears that x_pan should be HWxd_pan?
* l148: post-processing is unclear.

Suggestions

S1. it may be interesting to mention the following related work:
* Fully Convolutional Networks for Panoptic Segmentation with Point-Based Supervision. TPAMI 2023.
* Panoptic SwiftNet: Pyramidal Fusion for Real-Time Panoptic Segmentation. Remote Sensing. 2023.
* Panoptic, Instance and Semantic Relations: A Relational Context Encoder to Enhance Panoptic Segmentation. CVPR 2022.

**Questions:**

Q1. Can you disentangle the relative contribution of loss relaxation and local enhancement (cf Table 7)?


**Limitations:**

It would be interesting to discuss whether there is any benefit in conjunction with weaker (RN-18) and stronger (SWIN, ConvNext) backbones.

---

> ### Author Rebuttal · Authors · 2023-08-09
>
> We thank the reviewer for the constructive feedback. We will improve the writing of the final paper based on all reviewers' feedback. In the meantime, we hope the answer below can help improve the readability of the paper.
>
> *W1. (1) lines 161-181 requires non-linear reading efforts.*
>
> We guess this might be due to the reviewer finding it hard to read the paper while referring to Figure 2 since it is separated apart into two pages. To make the paper easier to read, we will put the text of L161-181 close to Figure 2 together on the same page.
>
> *W1. (2) Equations for $x\_{pan}$ and $x\_{sem}$ are missing*
>
> Thanks for the suggestion. $x\_{pan}$ and $x\_{sem}$ indeed start from the shared features. To make it clear, we plan to add a bird's-eye view figure which will better illustrate how $x\_{pan}$ and $x\_{sem}$ in Figure 2 are related to the overall architecture, e.g. kMaX-Deeplab or MaskFormer.
>
> *W1. (3) d_pan, d_sem, N_Q and N_C should be defined before use.*
>
> We kindly argue that we did not define $d_{pan}$, but use $N_Q$ since it represents the number of queries for the transformer decoder. Finally, we thank the reviewer for the suggestion. We will move all these notations currently provided in L166-169 to an earlier part of the paper, closer to where they are first used, i.e. line 164.
>
> ---
>
> *W2. A bird's-eye view figure is missing (example: Fig.2 in [10]).*
>
> Thank you for the thoughtful suggestion.
> Due to the original space constraint, we didn't include a bird's eye view figure.
> We plan to add a bird's-eye view figure which will better illustrate how $x\_{sem}$ and $x\_{pan}$ in Figure 2 are related to the overall architecture, e.g. kMaX-Deeplab or MaskFormer.
>
> Meanwhile, we would like to point out that Figure 2 showcases the ReMask process, while Figure 3 details ReClass. They are presented in separate figures because ReClass is solely related to the loss and does not alter the architecture. Regarding the entire process, it completely followed kMaX-DeepLab. We recognize that this may be challenging for those unfamiliar with mask transformers.
>
> ---
>
> *W3. (1) it appears that $x\_{pan}$ should be HWx $d\_{pan}$?*
>
> We thank the reviewer for the question. We kindly note that we have never used (or defined) $d\_{pan}$ in our manuscript and $x\_{pan} \in \mathbb{R}^{HW\times N\_Q}$ is defined in L161. This is due to the structure of masked transformers, where the number of panoptic masks is defined by the number of queries $N\_Q$. We also have defined another term $d\_{q}$ in L167-168.  I hope this will clarify this confusion.
>
> *W3. (2) Post-processing is unclear.*
>
> Thanks for the question. We completely followed the post-processing in kMaX-DeepLab [64] without any change. We will mention this technical detail in the final paper and make it clear to readers.
>
> ---
>
> *S1. Related work.*
>
> Thanks for the supplement. We will add and discuss all three papers in the revised paper.
>
> ---
>
> *Q1. Can you disentangle the relative contribution of loss relaxation and local enhancement (cf Table 7)?*
>
> This is a good question! We added another ablation study that removes the semantic masking (the concrete grey arrow right under "stop grad" in Figure 2).
> This would keep the semantic loss $\mathcal{L}\_{sem}$ for semantic relaxation but remove the local enhancement (semantic masking). The result is shown below and will be added to the final paper.
>
> | *w/* semantic masking?$^{[a]}$ | *w/* $\mathcal{L}\_{sem}$?$^{[b]}$ | *w/* ReClass? | PQ |
> | :--------------------: | :------------------------: | :-: | :- |
> | | | | 50.4 |
> | | &#x2611; | | 51.3 |
> | &#x2611; | &#x2611; | | 51.7 |
> | &#x2611; | &#x2611; | &#x2611; | 52.4 |
>
> The table above shows that the semantic relaxation would lead to a 0.9 increase in PQ; while with the semantic masking it would lead to an additional 0.4 PQ gain. The semantic masking can be linked to local enhancement as it would suppress the extreme false-positive predictions via a simple masking operation.
>
> ---
>
> *Q2. Results on R18 and stronger backbones (Swin/ConvNext)*
>
> We thank the reviewer for the suggestion, and kindly remind the reviewer that we have evaluated our method on weaker backbones that are even smaller than ResNet-18 (MobileNetV3-Small and MobileNetV3-Large in Table 1 and 8). We also reported the result for the ConvNeXt model in Table A2 of the supplementary material.
>
> ---
>
> $^{[a]}$: local enhancement
>
> $^{[b]}$: semantic loss relaxation

---

> > ### Comment · Reviewer_8sND · 2023-08-19
> >
> > Thank you for the feedback, the proposed changes improve the manuscript. My d_pan indeed corresponds to N_q. For some reason that was not completely clear to me while reading the manuscript.

---

> > > ### Author Response · Authors · 2023-08-21
> > > **Thank you for your kind feedback**
> > >
> > > We thank the reviewer for the recognition of our efforts. We will revise the paper accordingly and try our best to make it more readable for the readers.

---

### Official Review · Reviewer_5TWL · 2023-07-10

**Soundness:** 3 good
**Presentation:** 3 good
**Contribution:** 3 good
**Rating:** 6
**Confidence:** 4

**Summary:**

This paper introduces a novel mechanism called ReMaX to enhance the training of mask transformers for efficient panoptic segmentation, making it more accessible and practical. The authors observe that the high complexity of panoptic segmentation training objectives often results in imbalanced loss, leading to challenges in training end-to-end mask-transformer based architectures, particularly for efficient models.
In response to this challenge, the authors propose ReMaX, which incorporates relaxation techniques for mask predictions and class predictions during training for panoptic segmentation. Through these simple relaxation strategies, the model consistently improves its performance without incurring any additional computational cost during inference.
The effectiveness of ReMaX is demonstrated by integrating it into efficient backbones like MobileNetV3-Small. The proposed method achieves a new state-of-the-art record for efficient panoptic segmentation on benchmark datasets such as COCO, ADE20K, and Cityscapes.
The results showcase the significance of ReMaX in improving the performance of mask transformers and its potential for advancing the field of efficient panoptic segmentation.

**Strengths:**

By applying such simple techniques for relaxation to the state-of-the-art kMaX-DeepLab, ReMaX can train the network stably without any gradient-clipping operation under a learning rate that is over 10× greater than the baseline. Experimental results have show that the proposed method both boosts the training speed by 3×, and also leads to much better results for panoptic segmentation. ReMaX sets a new state-of-the-art record for efficient panoptic segmentation.

**Weaknesses:**

(1) Do you have the experimental results on test set for COCO, CityScapes and ADE20K?
(2) How do you balance Lpan and Lose? Are there any weights for these two losses?
(3) In Table 2, what is the result of using softmax as activation and with grad-clip?
(4) In Table 5, I'm wondering why removing the auxiliary semantic head will not lead to performance drop when using identity mapping.
(5) In Table 7, why does PQ drop a lot when you use ground-truth semantic masks for m_sem? What is the result of using ground-truth semantic masks with the stop-gradient operation?

**Questions:**

I'm positive about this paper. However I still have some concerns in the Weaknesses. I'll make the final decision after I see the response from the authors for those questions in Weaknesses.

**Limitations:**

I cannot find any limitations or potential negative societal impact which the authors list.

---

> ### Author Rebuttal · Authors · 2023-08-09
>
> We appreciate the reviewer's valuable feedback. We hope that the subsequent response will clarify the issues highlighted and contribute to an improved rating.
>
> *Q1. COCO/CityScapes/ADE20K test set*
>
>    Thank you for the suggestion. We intend to include test set results for these datasets in the finalized version. Regrettably, due to the shortage of training resources (GPUs) and time constraints, we are unable to present these results in the current rebuttal.
>
> ---
>
> *Q2. loss weights on $\mathcal{L}\_{pan}$ and $\mathcal{L}\_{sem}$.*
>
>    We have mentioned it in L246-247, and L307-308, respectively. We kindly note that the panoptic losses $\mathcal{L}_{pan}$ consist of multiple losses including PQ-style losses and mask-id loss.
>    We suppose the current inconsistency in loss names may lead to confusion for the audience to read, therefore, we would revise L246-247 to:
>
>    > The loss weight for $\mathcal{L}\_{sem}$ is 0.5 and that for $\mathcal{L}\_{pan}$ is set as the same with kMaX-DeepLab [64].
>
>    Then we would revise L307-308 to:
>
>    > The weights for the PQ-style loss (part of $\mathcal{L}\_{pan}$), auxiliary semantic loss ($\mathcal{L}_{sem}$), mask-id cross-entropy loss (part of $\mathcal{L}\_{pan}$), and instance discrimination loss are set to 3.0, 1.0, 0.3 and 1.0.
>
> ---
>
> *Q3. In Table 2, what is the result of using softmax as activation and with grad-clip?*
>
>    Softmax tends to produce fewer False Positives than Sigmoid because each pixel is limited to a single positive prediction, as outlined in lines L24-26 of the manuscript.
>    By contrast, Sigmoid permits each pixel to correlate with several mask predictions, which can result in a highly unbalanced loss during training.
>
>    When applying Softmax with grad-clip, the network converged too slowly (10+ points lower than the baseline) because of insufficient positive gradients; so we did not report it in Table 2.
>
> ---
>
> *Q4.  In Table 5, I'm wondering why removing the auxiliary semantic head will not lead to a performance drop when using identity mapping.*
>
>    We employed identity mapping to preserve the initial prediction, with the semantic branch serving solely as a relaxation mechanism. The overarching goal for panoptic segmentation remains unaltered.
>    It's important to highlight that only the **`first four stages`** utilize ReMask, not all mask decoders. Eliminating the semantic branch might influence the intermediate predictions, but it won't have a direct impact on the final prediction.
>    This may suggest that variations in intermediate predictions might not substantially influence the final performance.
>
> ---
>
> *Q5. In Table 7, why does PQ drop a lot when you use ground-truth semantic masks for $m\_{sem}$? What is the result of using ground-truth semantic masks with the stop-gradient operation?*
>
>    This is a good question. To make it clearer, using the ground-truth semantic masks (gt-masks) means that :
>
>    1. There is no semantic loss during training, which does not provide any relaxation for the training objective as semantic segmentation is a sub-task of panoptic segmentation.
>
>    2. All false-positive losses outside the gt-masks would be eliminated, indicating that the false-positive losses are still important. Removing most of them will lead the network to converge to sub-optimal.
>
>    3. ReMask here is to help eliminate the **`extreme`** false positive losses to prevent the loss distribution to be unbalanced (Figure A2 in supplementary material).
>
>    We would like to clarify that when using ground-truth masks for semantic masking, there would be no parameters in the semantic branch; therefore, there is no gradient to be stopped in this scenario.

---

> > ### Comment · Reviewer_5TWL · 2023-08-21
> > **Increase my rating to weak accept**
> >
> > Thanks for the author's response. It has addressed all of my concerns and I'll raise my rating to weak accept. Thank you.

---

### Decision · Program_Chairs · 2023-09-21

**Decision:**

Accept (poster)

**Comment:**

In this work, the authors propose two training techniques for panoptic segmentation, namely ReMask and ReClass, to address issues in panoptic segmentation training. The proposed techniques are simple and sweet, and are validated to be effective with good results. The work received unanimous recommendations on acceptance from the reviewers. The AC agrees with the reviewers and recommends acceptance. The authors are highly encouraged to incorporate the suggestions/results from the rebuttal discussions in their final version and revise accordingly.